# Integration of the Standalone 'Risk Assessment' Section in Project Level Environmental Impact Assessment Reports for Value Addition: An Indian Case Analysis

**Arjun Kumar A. Rathi**

Faculty of Planning and Public Policy, CEPT University, Ahmedabad 380059, India; drakarathi@gmail.com

**Abstract:** Both environmental impact assessment (EIA) and risk assessment (RA) instituted some 50 years ago are interdisciplinary and decision-support tools and have analogies in their procedural steps. Environmental risk assessment could be employed as such or as complementary to EIA for environmental management. This study aims to examine how RA is dealt with in the EIA reports of greenfield projects. The investigation reveals that RA is treated as a standalone exercise and too inadequately in EIA studies. There is a lack of well-defined criteria and methodologies for RA in different contexts, evaluation of prediction uncertainties, residual risks, assimilating RA in EIA, regulatory framework to strengthen RA integration in EIA, objective review of RA by the competent authority, and EIA follow-up. Unambiguous terms of reference are proposed for RA in EIA under the prevailing regulation for immediate implementation. The duration and cost of preparing and reviewing EIA reports integrated with RA would increase but there would be more value addition to the EIA studies. Comprehensive EIA regulation, RA-related scoping, and institutional capacity building could help promote such integration that is crucial for assessing industrial and other anthropogenic calamities at the project development stage.

**Keywords:** comprehensive EIA regulation; comprehensive RA-related scoping; environmental risk assessment; risk assessment; RA integration into EIA; India; uncertainties in EIA

## 1. Introduction

Environmental impact assessment (EIA) and risk assessment (RA) are the most prominent and widely employed environmental techniques for environmentally compatible development. While EIA essentially involves the identification and evaluation of the environmental and social impacts of a proposed project, RA involves the identification of hazards and the evaluation of the associated risks. EIA was instituted in the 1970s, with a primary focus on impacts on the physical, ecological, and social components of the environment. RA is one of the oldest decision-making tools, and its application was initiated in the estimation of human health risks [1] from chemicals in the water in the early 1980s, and the European Union developed RA protocols relating to the environmental impacts in 1992.

RA considers complexity, uncertainty, and ambiguity [2]. The core elements of RA were identified by [3] as (a) risk identification—establishing a cause–effect link of risks; (b) exposure and/or vulnerability prediction—modeling diffusion, exposure, and effects on receptors; (c) risk estimation—determining the strength of a cause–effect link. These components are similar to those of EIA, viz. impact identification, prediction, and assessment/evaluation, respectively. There are analogies in RA and EIA process steps [4], viz. hazard identification in RA is equivalent to screening and scoping in EIA, risk estimation to impact prediction, risk evaluation to the determination of the significance of the impact, and risk management, specifically, risk reduction and control to impact mitigation. In the framework, both EIA and RA have evolved as analogous and at times overlapping

procedures with well-developed methodologies [5] to provide an adequate basis to facilitate decision making. As such, the framework of RA and EIA is similar [6], involving the prediction of future impacts of the proposed activities and aiming to aid the decision-making process for significance, magnitude, and type of impacts, acceptability of risk, and mitigation measures.

The science of RA is increasingly complex. RA, primarily developed as a process to analyze risks associated with different types of development [7], comprises characterization of the nature and magnitude of health risks to humans and ecological receptors from chemical contaminants and attributes such as air and water pollution, hazardous substances, hazardous wastes, and toxic exposure present in the environment. Environmental RA (ERA), consisting of human health RA and ecological RA, contributes to achieving the goals of sustainable development [8]. The overviews of the growing area of ERA and environmental risk management (ERM) are well-documented [9,10]. Like EIA, ERA provides technical support to evaluate prevalent environmental settings and anticipate future settings under the selected scenarios. Effective ERM [11] is a continuous, participatory, and forward-looking process that foresees and estimates likely impacts to plan and manage project-related activities by mitigating adverse impacts. RA can be considered a complement of EIA [12], and ERA could be employed as such or as complementary to EIA to aid environmental decisions. In quantitative RA, ecosystem RA strategies [13] are evaluated to ascertain that risks associated with the exposure of ecosystems to chemical contaminants remain within limits prescribed in the applicable regulatory standards. However, reservations are expressed about the use of RA in practice [14]. Descriptive EIA methods continue to be used, given the limitations in terms of well-established quantitative systems [15,16] in understanding the risks posed by the proposed activities. Given that RA is concerned more with well-defined regulatory issues and uses formal quantitative analysis of the probability of specific undesired events, it is more focused but much narrower than EIA [17]. RA can be used in formulating risk ranking [18] of projects/activities by prioritizing the critical hazards which could result in the worst health-related consequences [19] and estimating the probability of suffering. Thus, prioritization along with specific mitigative actions and contingency plans should be described in the environmental management program (EMPg) chapter [20] in EIA reports.

## 2. Risk Assessment in EIA

EIA has firm roots in legislation and regulatory frameworks, but there is no strong historical tradition of integration of RA approaches into environmental policy and legislation [21]. EIA guidelines are prescribed [22] to include RA in EIA reports with details on the context of the qualitative RA, methodology, and approach followed, results of the qualitative RA in different lifecycle phases of the proposed project, and quantitative RA for the identified significant residual risk. Abnormal risks could be addressed in EIA [23] by answering the questions: (a) What can go wrong with a project? (b) What adverse consequences might occur to human health and the environment? (c) What is the range of magnitude of adverse consequences? (d) How likely are these consequences? While (a) and (b), and the qualitative indication to (c) to some extent, are addressed in a typical EIA, all these questions, notably (c) and (d) along with quantification, are addressed in a typical ERA. RA helps in the assessment of specific impacts that are not easy to predict due to the high degree of uncertainty involved [24]. ERA in EIA is essentially an application of RA methods and RM techniques to the associated ecological, social, and economic issues [25,26] in aiding scoping of scientific studies, prioritization of impact assessment for managerial intervention, integrated assessment of environmental, social, and economic aspects, and management of uncertainty. Thus, while environmental impact predictions of EIA are deterministic, uncertainty is treated unambiguously in RA [21]. Further, the inclusion of RA in EIA has some possible advantages [10], viz. "(a) the encouragement for integrated thinking by interdisciplinary teams conducting EIA studies; (b) the opportunity to focus attention on risk reduction activities such as waste minimization, pollution pre-

vention, and mitigation actions; and (c) the inclusion of emphases on emergency response measures in the event of accidents and associated environmental perturbations".

There is increasing focus in EIA on integrative approaches [27] to make it more effective. The need for integration of RA and EIA arises [28] because all the impacts do not get assessed adequately in EIA. Future developments need to be anticipated and planned for under dynamic conditions [29]. The term "uncertainty", used by some researchers needs to be broadened to circumscribe the values and beliefs that affect environmental assessment (EA) in intricate and vague conditions [30,31]. Risk analysis in EIAs is investigated by several researchers [32,33]. The European Union Directive on EIA is deeply rooted in a risk-based concept, and several risk-related terms, viz. "magnitude and complexity of the impact", "probability of the impact", and "duration, frequency, and reversibility of the impact" are defined. A framework for RA in the broader context of EA based on a modified version of Health Canada's model for human health RA and RM was proposed [34]. The framework proposed for the EIA of chemicals [35] in the context of registration, evaluation, authorization, and restriction of chemicals regulation affirms that even with limited data it is possible to move from risk indicators to impact indicators that are more relevant for the analysis of socio-economic aspects. Guidelines were given long ago to conduct RA as a part of EIA [23,36], and its integration into project development and implementation processes. The components of RA in EIA reports for new chemical industry projects are reported [37]. Even though risk-based EA is considered to be an iterative process with some degree of inherent uncertainty, the EIA process could be improved by using risk analysis approaches [28] to investigate and evaluate environmental impacts. The wider use of risk-based approaches [38] is acknowledged as potentially helpful in defining the environmental risks more precisely and enabling focus on the key issues in environmental management including monitoring. Models were proposed [39–41] to integrate risk analysis into the EIA and combine social impact assessment and risk impact assessment [42] to evaluate the social impacts of risky projects as well as natural hazards and management of disasters.

EA can also help prevent the basic causes of disaster risk and manage risk [43,44]. This role of EA can be strengthened further by specifically combining disaster risk-related aspects [45,46] into the development planning through EA, and integrating EA into the pre-disaster planning of post-disaster decision making [47] to prevent disaster recurrence in the post-disaster period [48]. EIAs need to consider significant hazards, possible sources, and environmental issues that could trigger or aggravate potential disasters and impacts of conflicts and disasters and suggest measures for the reduction of disasters by meaningfully using RA in EIA [49]. Loss prevention RA is generally carried out for potential accidents in manufacturing and energy sector projects as per the regulatory requirement of specific legislation such as the Seveso II Directive in Europe but not of EIA. Given that RA literature is generally dominated by engineering resilience, without taking into account ecosystems existing in multiple regimes, a conceptual model was proposed [50] for resilience-based risk and impact assessment. A general methodology to apply risk analysis for the impact assessment of construction activities is reported [51] and an approach is proposed to analyze and evaluate risks based on predetermined, defined, and objectively justified indicators, and criteria to determine the probabilities and consequences of stressors/impacts on the environment and health within the EIA, given that the risk-based approach has several potential advantages [16] including better prediction and recognition of cause and effect, sensitivity analysis, continual learning, and optimal resource allocation. Risk-based techniques using a conservative approach in terms of the likelihood of occurrence of credible worst-case consequence scenarios to assess potential impacts from project activities and gather reasonable outcomes were adopted [52]. EIA is generally used to assess potentially severe or irreversible consequences of development projects but for highly uncertain impacts, objective and data-driven information is required. There are a few formal quantitative systems to estimate the risks posed by construction and development projects requiring EIA [53,54] even though risk-based approaches [15] are well-developed for several related environmental fields. For a structured risk management framework for the assessment

and compliance stages of the EIA process in Australia, the Bayesian Belief Network was proposed as a risk model to predict overall EIA risk. The generic structure of the EIA process prescribed by the Indian EIA regulation [55] necessitates a RA section as additional studies in the EIA reports of certain projects.

It may be observed from the above that much of the research in EIA is focused on the theory, practice, and review of EIA and the application of descriptive EIA methods and less attention has been given to methods to aid prediction in real systems and development projects for the integration of RA into EIA. This is also evident from the fact that publications on topics related to RA in EIA were not found in the leading journals, viz. *Impact Assessment and Project Appraisal*, *Journal of Environmental Assessment and Policy Management*, *Journal of Environmental Planning and Management*, and *Integrated Environmental Assessment and Management* in the period 2015–2020, with the *Environmental Impact Assessment Review* being the exception [28,29,51].

## 3. RA in EIA: Indian Scenario

### 3.1. Objectives of the Study

The literature reviewed in Section 2 above reveals the importance and advantages of integrating risk analysis into EIA and adopting risk-based approaches/frameworks for EIA. As observed, empirical studies are not reported on the integration of RA in EIA practice. Given this, an investigation was undertaken to understand how RA is dealt with in the Indian EIA system by seeking answers to the following research questions: (a) How are RA-related aspects treated in the pre-EIA stages of EIA reports? (b) How effective is the RA-related coverage in the EIA reports? (c) To what extent are RA-related aspects integrated into EIA reports? (d) How is RA treated in EIA? (e) How is RA treated in EIA follow-up? (f) How comprehensive are the terms of reference (TOR) prescribed to incorporate RA-related aspects in EIA?

### 3.2. Methodology

To seek answers to the above research questions, 22% of the EIA reports, numbering 27 were selected randomly from 125 EIA reports [56] of green-field projects in manufacturing and energy sectors, viz. active pharmaceutical ingredients and intermediates (7), dyes and intermediates (1), synthetic resins (2), pesticides (3), chemical fertilizers (2), distillery (3), integrated steel (3), cement (2), and coal-based thermal power (4) that were granted environmental clearance [56] over eighteen months while ensuring that these were prepared by different EIA consultants. Given the toxic, hazardous, and inflatable characteristics of the chemicals used or produced in the first six types of the projects and the size, land requirement, and air pollutant emissions from the last three types of the projects, it is necessary to integrate risk assessment in EIA projects for well-informed decision making. The applicable, as well as proposed, EIA regulations were examined for the risk-related provisions. The risk-related aspects covered in the EIA reports are assessed for each of the prescribed standard TOR [57], and the findings are presented in Tables 1–3 and for specific TOR that are prescribed for certain project categories in Tables 4 and 5. The selected EIA reports were thoroughly examined to assess how RA is treated in the EIA reports and how appropriately the TOR prescribed by competent authority are addressed, considering that EIA studies are generally undertaken based on the prescribed TOR. Given the small sample size, quantitative analysis is not attempted. This is the limitation of this study. The adequacy of RA-related TOR in the EIA reports is evaluated on a scale of 0–3: 0 being no context or out of context, 1 inadequate, 2 reasonably adequate, and 3 adequate. Based on the objective of each TOR, comprehensive, unambiguous, and easy-to-understand RA-related TOR are proposed to enable the EIA team to address these in the EIA reports comprehensively and the EIA reviewers to appraise these thoroughly.

In the context of the research questions posed to analyze how the RA is dealt with in the EIA reports, the methodological approach [58–60] is adopted for the completeness criterion for the major stages of the EIA process, viz. screening, scoping, preparation of the

EIA report, decision making, and EIA follow-up, given that this methodology is simple to use and understand besides being universal and versatile.

## 4. Findings

Based on the prescribed TOR, the RA study is undertaken and the RA section is included in the EIA reports of development projects. To reduce delays, remove arbitrariness and make the EIA process transparent, standard TOR [57] are specified in India for different project sectors so that the EIA study could be taken up immediately after the successful online registration of the proposal to seek environmental approval. It is observed that the risk-related aspects are confined to the RA section which is considered to be a part of the additional studies chapter in the EIA reports. It is found that the RA section does not use information from other chapters and the outcome of this section is not used in other chapters of the EIA report. Thus, the RA section is treated as a "standalone" in an EIA report. Moreover, the EIA review and hence the decision-making also seem to be overlooking this section. The TORs are briefly described below and findings on how each TOR related to RA, occupational health, and safety is addressed in the EIA reports are presented in Tables 1–5.

**Table 1.** Standard TOR1.

| S. No. | Project | | EIA Reports | |
| | Sector | No. of Projects | Coverage in the RA Section | No. of EIA Reports |
|---|---|---|---|---|
| 1 | Chemical fertilizers | 2 | Conceptual RA. | 1 |
| | | | Tabulation of potential hazards. | 1 |
| 2 | Distillery | 3 | Conceptual hazard identification, RA, and methodologies. | |
| | | | Conceptual RA methodologies at length. | 1 |
| | | | Identification of hazards, corresponding risk, and universal mitigation measures; conceptual RA methodologies and fire radiation analysis; estimation of the consequence of releases using ALOHA software, and plotting of damage distance contours. | 1 |
| | | | | 1 |
| 3 | Integrated steel | 3 | Conceptual risk and risk evaluation, identification of hazard potential of different activities and risks. | 1 |
| | | | Identification of risk of different activities and universal risk mitigation measures, estimation of heat radiation effect distances under different MCA scenarios using ALOHA software. | 1 |
| | | | Identification of hazards of different activities, tabulation of different MCA scenarios, and heat radiation effect distances. | 1 |
| 4 | Pesticides | 3 | Concepts of hazards. | 2 |
| | | | Identification of hazards. | 1 |
| 5 | Synthetic organic chemicals | 10 | Conceptual hazard analysis, RA, and consequence analysis. | 3 |
| | | | Conceptual RA. | 3 |
| | | | Plant area-wise risk identification, generic precautionary and mitigation measures. | 1 |
| | | | Generalized hazards, causes, and risks. | 1 |
| | | | Not addressed. | 2 |

**Table 2.** Standard TOR2.

| S. No. | Project | | EIA Reports | |
| --- | --- | --- | --- | --- |
| | Sector | No. of Projects | Coverage in the RA Section | No. of EIA Reports |
| 1 | Cement | 2 | Conceptual RA, generic emergency preparedness, and DMP. | 1 |
| | | | Typical hazards, generic emergency preparedness, and DMP. | 1 |
| 2 | Chemical fertilizers | 2 | Generic emergency preparedness and elaborate DMP. | 2 |
| 3 | Distillery | 3 | Generic emergency preparedness and elaborate DMP. | 2 |
| | | | Elaborate emergency preparedness and DMP. | 1 |
| 4 | Integrated steel | 3 | Generic emergency preparedness and DMP. | 2 |
| | | | Generic risk management measures, emergency preparedness, and DMP. | 1 |
| 5 | Pesticides | 3 | Generic emergency preparedness and DMP. | 3 |
| 6 | Synthetic organic chemicals | 10 | Generic emergency preparedness and DMP. | 4 |
| | | | Generic emergency preparedness and elaborate DMP. | 3 |
| | | | Concepts of disaster management. | 1 |
| | | | Generic emergency plan. | 1 |
| | | | Not addressed. | 1 |
| 7 | Thermal power plants | 4 | Conceptual preliminary hazard analysis at length, usual types of hazards; generic emergency preparedness and DMP at length; fire-fighting system. | 1 |
| | | | Concepts of RA; estimation of fire and impact distances using ALOHA software; risk mitigation measures; fire-fighting system; standard safety measures. | 1 |
| | | | Identification of hazards associated with plant activities; RA and consequence analysis of fire and releases using ALOHA software, and risk mitigation measures; usual emergency | 1 |
| | | | Identification of hazards associated with plant activities; conceptual RA, estimation of heat radiation effect distances of fire under different MCA scenarios using RADN consequence software; mention of coal dust explosion; standard emergency preparedness and DMP. | 1 |

**Table 3.** Standard TOR3.

| S. No. | Project | | EIA Reports | |
| --- | --- | --- | --- | --- |
| | Sector | No. of Projects | Coverage in the RA Section | No. of EIA Reports |
| 1 | Cement | 2 | Generic OH&S hazards along with usual control measures. | 1 |
| | | | Generic OH&S hazards along with usual control measures, description of PPE, OH budget for 3 years. | 1 |

**Table 3.** *Cont.*

| S. No. | Project | | EIA Reports | |
| --- | --- | --- | --- | --- |
| | **Sector** | **No. of Projects** | **Coverage in the RA Section** | **No. of EIA Reports** |
| 2 | Chemical fertilizers | 2 | Properties of specific hazardous chemicals and their physiological effects on human beings. | 1 |
| | | | Generic OH&S measures. | 1 |
| 3 | Distillery | 3 | Generic OH&S measures, and PPE. | 2 |
| | | | Generic OH&S measures. | 1 |
| 4 | Integrated steel | 3 | Identification of OH&S hazards and control measures, and PPE. | 1 |
| | | | Generic OH&S measures, and PPE. | 1 |
| | | | Generic OH&S measures. | 1 |
| 5 | Pesticides | 3 | Generic OH&S measures. | 3 |
| 6 | Synthetic organic chemicals | 10 | Generic OH&S measures. | 5 |
| | | | Generic aspects of industrial OH. | 1 |
| | | | Occupational health center. | 1 |
| | | | Not addressed. | 3 |
| 7 | Thermal power plants | 4 | Generic OH&S measures. | 4 |

**Table 4.** Specific TOR4.

| S. No. | Project | | EIA Reports | |
| --- | --- | --- | --- | --- |
| | **Sector** | **No. of Projects** | **Coverage in the RA Section** | **No. of EIA Reports** |
| 1 | Pesticides | 3 | Not addressed. | 3 |
| 2 | Synthetic organic chemicals | 10 | Generic aspects of OH&S. | 5 |
| | | | Generic aspects of toxic management, medical surveillance. | 1 |
| | | | Material safety data sheets for chemicals. | 1 |
| | | | Not addressed. | 3 |

**Table 5.** Specific TOR5.

| S. No. | Project | | EIA Reports | |
| --- | --- | --- | --- | --- |
| | **Sector** | **No. of Projects** | **Coverage in the RA Section** | **No. of EIA Reports** |
| 1 | Chemical fertilizers | 2 | Risk evaluation and consequence analysis using PHAST-RISK Micro software for failure scenarios and failure frequencies, estimation of hazard distances of thermal radiation under different weather conditions, plotting of risk contours, and generalized recommendations. | 1 |
| | | | Inventory of storages, threat zones estimation under different MCA scenarios and weather conditions using PHAST software, plotting of risk contours; standard precautionary measures for fire and explosion. | 1 |

**Table 5.** *Cont.*

| S. No. | Project | | EIA Reports | |
| | Sector | No. of Projects | Coverage in the RA Section | No. of EIA Reports |
|---|---|---|---|---|
| 2 | Pesticides | 3 | Properties of chemicals and details of storages, the concept of QRA, estimation of impact zones under different MCA scenarios and weather conditions using PHAST Micro software, plotting of risk contours, generally applied good safety practices. | 1 |
| | | | Properties and inventory of storages, estimation of impact zones under different MCA scenarios and weather conditions using ALOHA software, risk contours, risk reduction, and control measures, generally applied good safety practices. | 1 |
| | | | Conceptual QRA at length, details of storages, estimation of toxic impact zones under different MCA scenarios and weather conditions using ALOHA software, generally applied good safety practices. | 1 |
| 3 | Synthetic organic chemicals | 10 | Details of storages, hazardous properties with NFPA codes, consequence analysis for failure scenarios of specific chemicals and estimation of threat zones using ALOHA software; and typical good safety practices and precautions. | 2 |
| | | | Inventory of chemicals, mode of storage, hazardous properties with NFPA rating; typical good safety practices; effect and consequence analysis of the release of chemicals for possible accident scenarios (without any estimation). | 1 |
| | | | Properties of chemicals and details of storages with NFPA codes, concepts of fire and explosion indices, consequence analysis for failure scenarios of specific chemicals, and estimation of toxic threat zones using ALOHA software. | 1 |
| | | | Inventory of chemicals, mode of storage, hazardous properties. | 1 |
| | | | Typical good safety practices; and estimation of threat zones using ALOHA software for the release of chemicals. | 1 |
| | | | Consequences and typical good safety practices. | 1 |
| | | | Conceptual RA, properties of chemicals and details of storages, estimation of threat zones using ALOHA software, and typical good safety practices. | 1 |
| | | | Properties of chemicals and details of storage, the concept of QRA, estimation of impact zones using PHAST Micro software, risk contours, and typical good safety practices. | 1 |
| | | | Not addressed. | 1 |

### 4.1. Standard TOR1—Hazard Identification and Details of the Proposed Safety Systems

TOR1 prescribes the incorporation of hazard identification and details of proposed safety systems in the project description chapter but these are usually discussed in the RA section in the EIA reports. The terms "hazard" and "risk" are generally used interchangeably in most of the EIA reports giving an impression of a lack of clarity that a hazard [61] is an inherent physical or chemical characteristic that has the potential to cause harm to the people, property, or environment whereas risk [62] is usually defined as a combination of how often an event can occur (probability of occurrence) and how dangerous it is when it does occur (severity of consequence).

*4.2. Standard TOR2—Onsite and Offsite Disaster (Natural and Man-Made) Preparedness and Emergency Management Plan Including Risk Assessment and Damage Control, and Disaster Management Plan (DMP) Linked with District Disaster Management Plan*

The disaster management plan (DMP) is required to be incorporated in EMPg, but it forms part of the RA section and is not even referred to in the EMPg chapter in the EIA reports. Coal-dust explosion is mentioned as a potential risk in one of the four EIA reports of coal-based thermal power projects, albeit without any details. Fire and explosion in the coal yard and coal pulverizing are not even considered potential risks in the EIA reports of thermal power projects though pool-fire scenarios due to leakage from auxiliary liquid fuel storage tanks are developed and impact distances estimated using the software.

*4.3. Standard TOR3—Plan and Fund Allocation to Ensure Occupational Health and Safety (OH&S) of Contract and Casual Workers*

The generic aspects of OH&S are described and most of the EIA reports consider OH aspects in the operation phase only. Personal protective equipment (PPE) is described at length as control measures for OH&S.

*4.4. Specific TOR4—Arrangements for Ensuring the Health and Safety of Workers Engaged in the Handling of Toxic Materials*

This aspect is not covered in most of the EIA reports. Further, no report used RA to prioritize critical hazards resulting in the worst health-related consequences [19]. A specific plan to ensure OH&S of workmen from hazardous chemicals and solvents proposed to be handled, used, or produced is not described. Generic control measures for dealing with OH&S are mentioned in most of the EIA reports overlooking the predicted concentration contours of toxic emissions for leakage scenarios. PPE is generally considered a standard OH&S mitigation measure, overlooking the typical hierarchy of mitigation of impact avoidance at the top.

*4.5. Specific TOR5—Risk Assessment (RA) for Storage and Handling of Hazardous Chemicals/Solvents, and Action Plan for Safety System*

Inventory of raw materials, solvents and products, mode of storage, hazardous properties, and National Fire Protection Association rating are tabulated in many EIA reports, and standard safety practices for handling, storage, and transportation are generally described. Consequence modeling using software to estimate heat radiation effect distances due to pool-fire from solvents and liquid fuels under different maximum credible accident scenarios and weather conditions is described, impact distances tabulated, and risk contours plotted in several EIA reports, albeit without any discussion.

Further, several similarities are observed in the RA section in the EIA reports, even though the reports are prepared by different EIA consultants. This could be attributed to the accredited risk and hazards experts who are permitted to be empaneled with five EIA consulting organizations [63], and they appear to have standardized their respective templates to address the standard and specific TOR on RA. Moreover, the EIA reports submitted for environmental approval are available in the public domain.

**5. Discussion**

How RA is dealt with in the Indian EIA system is discussed below using the methodological approach [58] in the context of the research questions raised in Section 3.1 above.

(a) How are the RA-related aspects treated in the pre-EIA stages, viz. screening and scoping?

The schedule to the EIA regulation [55] classifies development projects covered under mandatory EIA into different categories based on size thresholds. The project sectors include extraction of minerals, manufacturing and processing, energy, infrastructure, and construction. The screening of projects for RA studies is done based on the type of project and project-wise standard and specific TOR [57] that are specified, irrespective of the project size or location. The draft regulation [64] exempts micro, small, and medium-sized

projects from the provisions of the EIA regulation, irrespective of their type and location. Standard TOR for RA consider the contexts of occupational health and disaster risk but overlook the contexts of the project size and location, technology and practices, health, ecological, socioeconomic, and cultural components of the environment, uncertainties in impact assessment, typology of significant impacts, and residual impacts, etc., that are essential for practical risk-based decision making to help ensure protect public health, and the environment [65] and integrate the RA into EIA reports. The ex ante EIA regulatory framework, which itself lacks robustness [19,66], tends to defeat the basic objective of the EIA process [67] by prescribing standard TOR rather than the project and location-specific TOR for each proposal. Given that the impacts and risks of different projects/activities are different [4], comprehensive scoping for RA in EIA is crucial for each proposal. For example, (a) RA for mining projects needs to consider the contexts of socioeconomic, cultural, and ecology due to impacts on land use/land cover, dust emissions, wastewater discharge, sediment transport, transportation, etc.; (b) infrastructure projects of structural failures and socioeconomic, cultural, and ecology impacts due to effect on land use/land cover, air emissions, noise, accidents, oil spills, etc.; and (c) manufacturing projects of equipment failure, air emissions, wastewater discharge, hazardous waste disposal, fire and explosion on health, ecology and social components of the environment under normal, abnormal, and non-operating scenarios, etc.

Thus, the screening, as well as the scoping stages of the RA process need strengthening to overcome inadequacies through regulatory provisions for a risk-based approach [15,16] for EIA by defining criteria to facilitate RA and correlate the proposed activities with the events that may have the potential to cause injury and detrimental consequences [68].

(b)     How effective is the RA-related coverage in the EIA reports?

Even where damage/impact distances are estimated and iso-risk contours are plotted for heat radiation, and/or concentrations of hazardous/toxic substances with the help of software, the extent of risk to susceptible receptors in the predicted impact zones under different scenarios is not covered. Generic emergency preparedness plans without correlating risks with specific risk mitigation and damage control actions add little value to the EIA reports. A lack of systematic hazard identification, use of methodologies for RA, and specific safety systems for risk mitigation in EIA reports is evident in spite of the accredited functional area expert on risk and hazards [63] in the EIA team. OH&S-related aspects of workmen are addressed in a generic manner in most EIA reports, possibly with an understanding that proposing any specific OH&S measures are likely to be inconsequential as these aspects get little weightage in the EIA review. The most probable reason for the inadequate addressing of the TOR on OH&S is that these aspects are governed by the Factories Act, which is administered by another competent authority, the Industrial Safety and Health Directorate at the state level. This overlap could be resolved through integrated regulation.

In the absence of methodical RA based on the activities and processes and standard methodologies to identify hazards and analyze the associated risks quantitatively or qualitatively using a risk matrix [66,69,70], the RA and onsite and offsite emergency preparedness plans or DMP are mere procedural formality devoid of objectivity. EIAs do not consider the outcome of the risks becoming reality, i.e., severity of the impact on the environment [71], given that risks are generally associated with the probability of the occurrence of environmental impact-causing events that have implications on biophysical, social, or economic components of the environment. Emergency response actions in the event of accidents and concomitant environmental variation [72] are also not reflected in the EIA reports. Like the topmost priority given to "avoidance" in a typical environmental mitigation hierarchy, there is a need to evaluate the root causes of well-established risks under the most credible accident scenarios. This study also confirms the limitations of the application of the effective ERA methods in EIA [26]: "several objectives of impact assessment, viz. objective determination of whether a risk is acceptable, evaluation of compliance with legislation

and policy as part of the EIA process, cumulative and synergistic impacts, etc. do not get addressed".

(c)    To what extent are the RA-related aspects integrated into the EIA reports?

None of the approaches [39,40,73] are employed to integrate risk analysis into the EIA and the RA section is presented as a standalone section, having no linkages with any other chapters of the EIA report including the EMPg. Such a 'silo-based approach' to treating RA in EIA reports [21] can never help serve the desired objective of encouraging integrated thinking by the multidisciplinary EIA teams. Charles Kelly, Co-Chairman, IAIA disaster/conflict section, opined in his email of 20 February 2021 to me: "So far, we have not seen the real integration of RA, as in the form of disaster RA, into the EIA process broadly. Note also that there is something of a semantic challenge. Impact assessment, as in an EIA, looks at the possible impacts of a proposed project. RA looks at the risk created by the intersection of society and the environment, that is, it is much broader. The conventional impact assessment is specific to an action to take place, while RA is general to any actions that might occur where a hazard is present". The current practice of having a standalone RA section in the EIA report is merely a tick-box outlook, a formality to fulfill the procedural requirement of the EIA framework, and an unproductive exercise without serving the intended objective is confirmed by this study too.

(d)    How is RA treated in the EIA review?

The EIA review involves the assessment of the quality, completeness, and adequacy of the information provided in the EIA report to make an overall judgment on its acceptability [74]. The evaluation of the adequacy of compliance with the TOR related to RA, and occupational health and safety aspects, summarized in Table 6, reveals that compliance with the TOR is low except for the TOR on RA for storage and handling of solvents and liquid fuels for which impact distances are predicted using software in many EIA reports. Granting environmental approvals despite inadequate compliance with the TOR gives the impression that the EIA review is based on narrow considerations, disregarding the completeness and quality of RA and its linkage with other chapters in the EIA report, especially the EMPg. It also reflects on the treatment given by EIA consultants to RA in EIA reports and a lack of regulatory provisions on the synthesis of RA into EIA. The lack of such vital information in EIA reports, especially for high-risk-potential projects impedes well-considered decision-making [75]: "theory is divorced from practice", and "the link between evaluation and decision-making is not enough".

**Table 6.** Evaluation of compliance to RA-related TOR in EIA reports.

| Parameter | * Evaluation Score | | | | |
|---|---|---|---|---|---|
| | TOR1 | TOR2 | TOR3 | TOR4 | TOR5 |
| Applicability to EIAs, no. | 21 | 27 | 27 | 13 | 15 |
| Score range | 0–3 | 0–2 | 0–2 | 0–1 | 0–3 |
| Mean score | 0.86 | 1.04 | 0.85 | 0.62 | 1.87 |
| Median value | 0 | 1 | 1 | 1 | 2 |
| Mode value | 0 | 1 | 1 | 1 | 2 |

* 0 poor, 1 inadequate, 2 reasonably adequate, 3 adequate.

For well-informed decision-making, it is necessary that there is an objective assessment of RA to serve as complementary to EIA to ascertain whether (a) the information obtained from air dispersion modeling for emissions of toxic substances such as ammonia, chlorine, or oleum, toxic concentration levels under maximum credible accident scenarios, and risk level assessed is unified and a comprehensive action program is incorporated in EMPg; (b) onsite emergency preparedness program, which should ideally form part of EMPg, is based on the outcome of RA studies to suggest appropriate safety systems, firefighting equipment, OH&S-related measures for heat radiation or toxic exposure in the vulnerable

zones, etc.; (c) risk mitigation programs to manage residual impacts, as well as uncertainties in impact predictions, are incorporated in EMPg; etc.

(e)     How is RA treated in the EIA follow-up?

The action plan on risk mitigation measures for human health, ecology, social and cultural impacts, loss prevention, disaster risk, emergency response preparedness, and specific disaster management are hardly described in the design for EIA follow-up besides ongoing mechanisms for monitoring, period audit, and review of risks to ensure the effectiveness of risk mitigation actions [20,24,70] in the EMPg in EIA reports. Because of the lack of coverage of these aspects, the risk mitigation and risk management-related aspects do not get followed up as a part of the post-project EIA follow-up which itself is weak [60,69,76]. It is, however, recognized that mandatory audits for the safety, health, and welfare of workmen and off-site emergency preparedness in manufacturing and energy sector projects are administered under the Factories Act at the state level and the Disaster Management Authority at the federal level responds to major calamities.

(f)     How comprehensive are the TOR prescribed to incorporate RA-related aspects in EIA?

The overlap and ambiguity of RA-related aspects in the standard TOR cause inadequate coverage of the RA in EIA. Comprehensive TOR could help EIA consultants and EIA reviewers develop clarity on the scope of the RA-related studies. Thus, the current TOR related to RA and emergency preparedness could be reformulated as follows to make these simple and comprehensive:

- Hazard identification of the substances proposed to be used and produced, processes, activities, and practices proposed to be employed in different lifecycle phases of the proposed project;
- RA of the identified hazards for human health, environmental health, and loss prevention under maximum credible accident scenarios and different weather conditions, suitably linked to air dispersion modeling carried out for the assessment of air quality in the impact assessment chapter;
- RA of residual impacts, uncertainty in impact predictions, scenarios of abnormal operations, happenings, or incidents, etc.;
- Details of specific safety systems/measures proposed for risk mitigation for the risks assessed, incorporated suitably in the environmental management program;
- Onsite emergency preparedness program, and offsite emergency preparedness program/disaster management program [77], considering RA at ii and iii above, incorporated suitably in environmental management program;
- Mechanism of periodic monitoring and management review of risks, adequacy of risk mitigation measures employed, and emergency preparedness program.

Likewise, the TOR related to occupational health and safety could be reformulated as follows:

- Identification of occupational health and safety-related issues in different lifecycle phases of the project;
- Specific comprehensive measures to address the risk assessed at ii and iii above to ensure the working personnel's occupational health and safety, including casual and contract workmen, suitably incorporated in the environmental management program.

## 6. Conclusions

Both the EIA and RA are interdisciplinary, complementary to each other, and decision-support tools with several similarities concerning (a) concepts, (b) objectives, (c) prediction of future consequences of the proposed projects by following similar procedures, (d) inherent uncertainties about the exact nature, frequency, and magnitude of consequences, (e) informing the decision-makers about the significance of adverse impacts or consequences, and impact or risk mitigation, (f) well-developed regulatory framework, etc. The treatment given to the RA section in the EIA reports at present hardly adds value to strengthening the EIA system because RA is considered a standalone exercise, more to

tick-box the checklist to fulfill the procedural requirements. The integration of RA into EIA is extremely important, given that RA facilitates the assessment of impacts that are not easy to predict due to the high degree of uncertainty involved albeit with some limitations. This, however, poses challenges since a lack of data may result in significant uncertainty.

For a more holistic EIA, it is necessary to (a) evolve a suitable regulatory framework for risk-based decision making; (b) design technically accurate RA and incorporate the appropriate and effective risk-management options for well-informed risk-based decision making; (c) build the required capacity of information, skills, suitable tools for determining risks under different conditions and scenarios, training and other resources to improve environmental decision making; and (d) make a system of stakeholder involvement to improve credibility and transparency in developing and applying RA into EIA, and enhance public confidence in risk management. In the broader context, risk governance [78] consisting of risk assessment, risk management, and risk communication is required with legal, institutional, social and economic contexts for risk evaluation. This could be done by using different technocratic or decisionistic models and transparent and inclusive governance models as integral to EIA.

Establishing the criteria for EIA quality and EIA review is not in the scope of this study. However, for the objectivity of the EIA review [79,80], risk analysis methods in EIA need to be prescribed [38,81] in the scoping for EIA reports addressing uncertainty and variability in EIA reports enhancing the quality of EIA studies [82,83]. The reframed TOR are proposed for RA in EIA to make these unambiguous and easy to comprehend for implementation. Well-formulated and comprehensive terms of reference on the risk-related aspects in the EIA for development projects, comprehensive regulation that integrates RA into EIA, and institutional capacity building on risk-based environmental assessment can help drive the integration of RA into EIA and improve the EIA system and EIA performance/effectiveness further. This may invite criticism that already burdened EIA will get overburdened, and the duration and cost of preparing and reviewing EIA will increase if a risk-based approach is incorporated into EIA. However, with proper scoping based on the significant risk potential and uncertainties, the benefits of a robust EIA, duly integrated with RA, are expected to outweigh the marginal cost increase. This study should also be of interest and relevance to other developing countries where economic growth is prioritized for poverty alleviation, employment generation, development and improvement of infrastructure, and EIAs are influenced by political and socio-economic conditions and constraints.

## 7. Way Forward

It needs to be recognized that environmental consequences form a link between the environmental aspects highlighted in the project description and the receptors portrayed in the description of the environment as illustrated in the impact significance flowchart [81]. Thus, the project description in EIA needs to capture all the project components, processes, operating scenarios, activities, associated ancillary, and support facilities over the project lifecycle, project implementation schedule, proposed technology, construction methodology, characterization, inventorization, and mode of storage and handling of different raw materials, intermediates, products, solvents, fuels, wastes, etc. These details should form the basis for hazard identification, RA, and hence proposing specific safety systems for risk mitigation, occupational health and safety of personnel, and emergency response program to meet any eventualities. The characterization of all the substances used and produced in manufacturing and energy-related projects, and proposed work practices should form the basis of exposure assessment, i.e., occupational health and safety hazards assessment without which a suitable program to help ensure the occupational health and safety of the working personnel cannot be worked out. Among others, the scoping for EIA reports should prescribe that both the safety systems and emergency response program, with a focus on receptors in the predicted vulnerable zones under the worst-case scenario, are elaborated in the environmental management program chapter in the EIA

report adequately to facilitate EIA follow-up [20,84] on the commencement of the project development and implementation.

The Indian EIA practice needs to pursue a risk-based approach [81,85], using objective and data-driven information to predict and attribute risk and assess potentially severe consequences and hence utilize the resources optimally [16]. To integrate RA into EIA and add significant value to strengthen the EIA system further, it is necessary to (a) properly define the concept of risk in the context of EIA, (b) develop detailed guidelines and methods for greater clarity on the expected role of RA in EIA [26], and (c) define and objectively justify indicators and criteria to determine the probabilities and consequences of impacts at the scoping stage in the EIA process for risk analysis in EIA in different contexts. A time-bound action plan to design technically accurate RA, taking into consideration risk-management options judiciously and formulating a suitable EIA framework for risk-based decision making would go a long way in adding value to EIA reports in which RA is integrated into EIA.

**Funding:** This research received no external funding.

**Institutional Review Board Statement:** Not applicable.

**Informed Consent Statement:** Not applicable.

**Data Availability Statement:** Data is included in tables in the manuscript.

**Conflicts of Interest:** The author declares no conflict of interest.

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
