# Peer review of "Integration of the Standalone ‘Risk Assessment’ Section in Project Level Environmental Impact Assessment Reports for Value Addition: An Indian Case Analysis"

_sustainability, doi:10.3390/su15032296_

Round 1

Reviewer 1 Report

This is not a research paper, and I do not think this should be published. There were no data presented. There is nothing original to contribute, and there is no advancement of science. This is similar to a report from a school class assignment.

Reviewer 2 Report

The author has embarked on investing a pressing but unresolved issue in the broad field of environmental risk assessment. Some of the questions related to the issue are a) is RA serving our purpose best by being a standalone decision-making tool? b) will RA serve environmental professionals and decision makers better if it is integrated with EIA? c) what is the best way to integrate RA into EIA? The research presented in the manuscript attempts to investigate some of these questions by examining case studies from India.

Abstract: the abstract is of appropriate length and incorporates adequate information about the major points of the article. It outlines the problem and summarises the main findings. I think ‘India’ could be a Keyword.

Introduction: The introduction introduces the concepts of EIA and RA and shows the similarities in their processes that creates the context for the question ‘is RA better off by being integrated into EIA?’ typo L78 “and i uses”

Risk Assessment in EIA: this is the literature review part of the manuscript. This section further elaborates on the relationship between ERA and EIA using some examples from developed countries, emphasises on research gap and develops arguments for integration of ERA and RA. The references seem appropriate given the vastness of literature on EIA and ERA and proliferation of related articles in the post-EIA era. I refer to the last para of this section, from reading this section it seems that the author had done a good review of relevant literature from related journals for 2015-2020. Some reflection on this would be useful. The author mentions lack of relevant articles in certain journals. I would like to see more specific information on this matter, e.g., what did the author find in EIAR? While an additional burden of work on the author, this would be very useful information and assist researchers in this field. If this has been outlined in the lit rev above, then repeating the references would suffice. Also, earlier in the same para the author made a general observation “Much of the research in EIA has focused on the theory, practice, and review of EIA and the application of descriptive EIA methods and less attention has been given to methods to aid prediction in real systems and development projects”. I would like to see some references here as well.

RA in EIA Indian scenario: This section focusses on Indian case study to examine how RA is treated in EIA reports. It lays out a few questions for investigation. Methodology is also outlined. There is room for improvement in this section. The author needs to describe the sectors and EISs they have selected for this research. What are these ‘green-field projects in manufacturing and energy sectors’? What is the relative risk nature of these sectors that require integration into EIA, that standalone RA is not adequate? The data collection methods need to be presented elaborately so readers can understand how the information in the discussion section was generated (see my comments under ‘Discussion’ below). Did the author interview EIA and RA professionals?  The author has adopted a well-established qualitative method of EIA evaluation and effectively used it in the Indian case. It is likely that the review of 22% EISs has generated significant amount of qualitative and quantitative information but the author has shared information in only 5 qualitative tables. I would like to see more facts and figures. Currently the ‘Findings’ section seem rather thin. It is said that 22% of EISs has been selected randomly from 125 EIAs. Readers would be interested to know the actual number here.

Discussion: This section is quite descriptive and general. It did not clearly establish its relationship with the findings section. Some clear statements to clarify the purpose of this section and its contents would be required. The author needs to clearly how this section is linked to the research questions and overall objective of the paper. I was a bit confused with the section (i, ii, iii, V ….).

Conclusion: this section seems adequate.

Way forward: Again, this section is descriptive and general. I see some repetitions here from discussion section. Also, this section is disproportionately large. The purpose of this section needs to be clear and the section to be brief.  

References: this section is adequate.

Overall: the manuscript has all the elements of a good article. But there is room for significant improvement in clarity and focus. I am particularly interested to see improvement in the methods section with clear articulation of activities involved in data collection, processing, and analysis. Also, the discussion section needs major rewriting for clarity. The section on ‘way forward’ needs justification. More facts and figures are needed to establish the validity of findings and conclusions.

Reviewer 3 Report

Dear author, thank you for submitting the manuscript to the journal Sustainability.

The subject could be relevant and appropriate for Sustainability.

However, the current version of the paper suffers from a number of weaknesses related to the empirical strategy used.

-   The abstract must contain the objective of the paper and the main findings.

-          It was impossible for me to identify the novelty of the paper. The paper should be revised to highlight novelties. Please consider that this lack of novelty starts with the Abstract, Introduction, and Conclusion. Besides, in the Introduction section, the author needs to present the aims and objectives as well as the scientific contribution of this work.

-          The word “Methodology” should be “methods” or “Materials and Methods”, the methodology is the study/analysis of methods and should only be used when addressing epistemologies/ontologies https://en.wiktionary.org/wiki/methodology#Usage_notes. This section needs to be reorganized and highly developed to clarify all steps of the proposed analysis. The author needs to include sufficient methodological details in the paper and elaborate on the produced results from the proposed methods. Some sections must be added and others need to be relocated and rewritten to make it clearer for the readers.

-          The author needs to understand that his research is based on scientific questions and should have a clear scientific contribution. The discussion section should be inserted by including a clear and concise analysis of all results presented in Section 4. It might be helpful to use more figures to present the results.

-          The references should be updated (1992?, 1993? 1994?...)

Overall, the manuscript is difficult to understand. The reader is not aware of the logical sequence of steps and the justification of the used procedures. It might be helpful to use more figures, tables, and charts to present the results.

 Best regards,

The reviewer

Round 2

Reviewer 2 Report

The author has now adequately responded to my initial review. 

Author Response

The reviewer has expressed satisfaction with the compliance with his initial review comments. There are no further comments.

As brought out earlier, the acronym TOR is used for terms of reference which is plural. However, the system considers it singular and shows spelling mistake.

Reviewer 3 Report

It is clear that the author has done a lot of work in order to improve the quality of the paper. However, there are still some aspects that need improvement in order for any reader could easily understand the paper.

Author Response

Reviewer 3 has acknowledged the additional work done by the author. He has not explicitly mentioned what more is expected from the author. As such, I am unable to respond.